# Cytoneme-mediated signaling essential for tumorigenesis

**Sol Fereres, Ryo Hatori, Makiko Hatori, Thomas B. Kornberg** *

Cardiovascular Research Institute, University of California, San Francisco, San Francisco, California, United States of America

* tkornberg@ucsf.edu

**Data Availability Statement:** All relevant data are within the manuscript and its Supporting Information files.

**Funding:** This work was funded by NIH fellowship 2T32HL007731 to S.F., 2T32HL007185 to R.H., and NIH grant CA205580 to T.B.K. The funders had

## Abstract

Communication between neoplastic cells and cells of their microenvironment is critical to cancer progression. To investigate the role of cytoneme-mediated signaling as a mechanism for distributing growth factor signaling proteins between tumor and tumor-associated cells, we analyzed EGFR and RET Drosophila tumor models and tested several genetic loss-of-function conditions that impair cytoneme-mediated signaling. *Neuroglian*, *capricious*, *Irk2*, *SCAR*, and *diaphanous* are genes that cytonemes require during normal development. Neuroglian and Capricious are cell adhesion proteins, Irk2 is a potassium channel, and SCAR and Diaphanous are actin-binding proteins, and the only process to which they are known to contribute jointly is cytoneme-mediated signaling. We observed that diminished function of any one of these genes suppressed tumor growth and increased organism survival. We also noted that EGFR-expressing tumor discs have abnormally extensive tracheation (respiratory tubes) and ectopically express Branchless (Bnl, a FGF) and FGFR. Bnl is a known inducer of tracheation that signals by a cytoneme-mediated process in other contexts, and we determined that exogenous over-expression of dominant negative FGFR suppressed tumor growth. Our results are consistent with the idea that cytonemes move signaling proteins between tumor and stromal cells and that cytoneme-mediated signaling is required for tumor growth and malignancy.

## Author summary

The growth of many types of tumors depend on productive interactions with stromal, non-tumor neighbors, and although there is evidence that tumor and stromal cells exchange signaling proteins and growth factors that they produce, the mechanism by which these proteins move between the signaling cells has not been investigated and is not known. Our previous work has shown that normal cells make transient chemical synapses at sites where specialized filopodia called cytonemes contact signaling partners, and in this work we explore the possibility that tumors use the same mechanism to communicate with stromal cells. We show that cytoneme-mediated signaling is essential for growth of Drosophila tumors that model human EGFR over-expression and RET-driven disease. Remarkably, inhibition of cytonemes cures flies of lethal tumors.

no role in the study design, data collection and analysis, decision to publish, or preparation of the manuscript.

**Competing interests:** The authors declare that no competing interests exist.

## Introduction

Human tumors include transformed tumor cells, blood vessels, immune response cells, and stromal cells that together with the extracellular matrix (ECM) constitute a "tumor microenvironment" [1]. The tumor microenvironment is essential for oncogenesis, cell survival, tumor progression, invasion and metastasis [2,3], and its stromal cells produce key drivers of tumorigenesis. Known drivers are growth factors (e.g. HGF, FGF, EGF, IGF-1, TGF-β and Wnts), cytokines (e.g. IL-6, SDF-1) and pro-angiogenic factors (e.g. VEGF). It is not known if these proteins function as autocrine, juxtacrine, or paracrine signals, nor is it known how they might move into or within the tumor microenvironment.

Studies of tumor models in Drosophila exploit the experimental attributes of the fly that provide uniquely powerful ways to investigate tumorigenesis [4]. We tested two models for the roles of cytonemes. Cytonemes are specialized, actin-based filopodia that extend between cells that produce and secrete signaling proteins and cells that receive them. The signaling proteins move along cytonemes and exchange at transient synapses that form where cytonemes contact target cells. These synapses are similar to neuronal synapses in constitution, structure and function [5–7], and are necessary for paracrine FGF/Bnl, BMP/Dpp, Hedgehog, Wnt/Wingless (Wnt/Wg), and Notch signaling during normal development of Drosophila epithelial tissues [5,7–9].

EGFR activating mutations are drivers of several types of human cancers [10]. However, elevated EGFR expression of wild type EGFR is not sufficient for tumorigenesis, and additional genetic changes are necessary, such as over-expression of Perlecan, a heparan sulfate proteoglycan (HSPG) component of the ECM [11]. In Drosophila, ectopic over-expression of Perlecan and EGFR in epithelial cells of the wing imaginal disc drives tumorigenesis [12]. Growth and metastasis of the epithelial cells require crosstalk with closely associated mesenchymal myoblasts, which also proliferate abnormally when Perlecan and EGFR are over-expressed in epithelial neighbors. The crosstalk includes BMP/Dpp signaling from the epithelial cells to the mesenchymal myoblasts [12].

The RET gene is the primary oncogenic driver for MEN2 (multiple endocrine neoplasia type 2) syndrome. MEN2 is characterized by several types of neoplastic transformations, including an aggressive thyroid cancer called medullary thyroid carcinoma (MTC). A fly model that overexpresses RET^MEN2 phenocopies aspects of the aberrant signaling in MEN2-related tumors, such as activation of the SRC signal transduction pathway, which promotes migration and metastasis of tumorigenic cells. The relevance of the fly model has been established by screens for small molecule suppressors of Drosophila tumors driven by RET^MEN2 over-expression. Several compounds that were identified are more effective than the drugs that are currently used for patients [13,14].

In the work presented here, we examined the role of cytoneme-mediated signaling in the EGFR-Pcn and RET^MEN2 models. Genetic inhibition of cytonemes by downregulation of five genes that were shown previously to be essential in cytoneme-mediated signaling, reduced tumor growth, and we describe genetic conditions that suppress lethality by as much as 60% in the EGFR-Pcn tumor and by as much as 30% in the RET^MEN2 tumor. Our results are consistent with the possibility that cytoneme-mediated signaling is necessary for tumor growth and that interfering with cytoneme-mediated tumor-stromal cell signaling might be a therapy for tumor suppression.

## Results

### Tumor cells and stromal cells extend cytonemes

Most of the wing imaginal disc is a columnar epithelium that will generate the wing and cuticle of the dorsal thorax of the adult fly. The disc also includes myoblasts that grow and spread

over much of the dorsal basal surface of the columnar epithelium; these mesenchymal cells will generate the flight muscles of the adult. Tracheal branches (respiratory tubes) also adjoin the basal surface of the columnar epithelium, and one branch, the transverse connective, sprouts a bud (the air sac primordium (ASP), Fig 1A) that initiates growth during the third instar period. The ASP is dependent on Dpp and Bnl signaling proteins produced by the wing disc [7]. The myoblasts relay Wg and Notch signaling between the disc and ASP [9]. Cytonemes mediate and are essential for the Dpp, Bnl, Wg, and Notch signaling [15].

To investigate whether cytonemes are also essential in tumorigenesis, we tested a cancer model that requires tumor-stroma interactions in which neoplastic transformation is driven by interactions between the wing disc epithelial cells and myoblasts [12]. Overexpression of wild type EGFR and Perlecan (*pcn*, a secreted heparan sulfate proteoglycan) in the columnar epithelium drives proliferation of the genetically modified epithelial cells, as well as their genetically wild type myoblast neighbors. Tumorigenesis depends on Dpp signaling from the epithelial cells to the myoblasts.

We first investigated if cytonemes are present in EGFR-Pcn overexpressing tumor cells. We induced the EGFR-Pcn tumor model (with *ap-Gal4*, an epithelial cell-specific driver) together with CD8:GFP, a membrane-tethered GFP protein (Fig 1C and 1D), and independently expressed membrane-tethered mCherry in the myoblasts (with *1151-lexA*, *lexO-mCherry-CAAX*; a myoblast-specific driver). In this system, the epithelial cell membranes are marked with GFP fluorescence and the myoblast membranes are marked with mCherry fluorescence. We observed that, as previously reported [12], the EGFR-Pcn tumor induces overgrowth and proliferation, producing multilayered masses of disorganized disc epithelial cells and myoblasts ([12], Fig 1D). Higher magnification imaging detected both epithelial cell cytonemes and myoblast cytonemes. Some of the cytonemes appear to extend between the tumor and mesenchymal populations (Fig 1E–1F'). These results show that tumor cells and tumor-associated cells extend cytoneme-like structures and are consistent with the possibility that cytonemes may facilitate signaling between these cell populations.

To monitor the tracheal branches that are associated with the tumorous discs, we induced the EGFR-Pcn tumor and labelled the trachea with membrane tethered GFP (with *LHG lex-O-CD2:GFP*, a tracheal-specific driver [16]). In the EGFR-Pcn tumor discs, the associated trachea were more extensive and branched than normal (Fig 1G and 1H). Their overgrowth was presumably a response to the disc tumor.

## Dpp localizes to tumor epithelial cell cytonemes

In normal development, Dpp produced by wing disc cells at the anterior/posterior compartment border is transported by cytonemes to target cells in both the wing disc and ASP, and cytoneme deficits caused by Capricious (Caps), Neuroglian (Nrg), or Diaphanous (Dia) loss-of-function lead to developmental defects [7]. In the EGFR-Pcn tumor model, Dpp signals from the genetically altered epithelial cells to drive myoblast expansion [12]. Dpp expression is upregulated in the epithelial cells (Fig 2A and 2A') and pMAD, the phosphorylated form of the Dpp signal transducer MAD, is enriched in the myoblasts (Fig 2A" and 2A'''). This indication of Dpp signal transduction in the myoblasts is consistent with previous results showing that Dpp signaling in this stromal compartment is required for tumor growth [12].

To investigate how Dpp moves in the EGFR-Pcn model, we used CRISPR mutagenesis to tag the endogenous Dpp protein with mCherry. mCherry was inserted at codon 465 as described in [17]. Homozygous Dpp:mCherry flies that lack a wild type *dpp* gene are viable and develop as wild type, indicating that the Dpp:mCherry chimera has normal function. We induced the EGFR-Pcn tumor in Dpp:mCherry flies, and labeled the EGFR-Pcn tumor cells

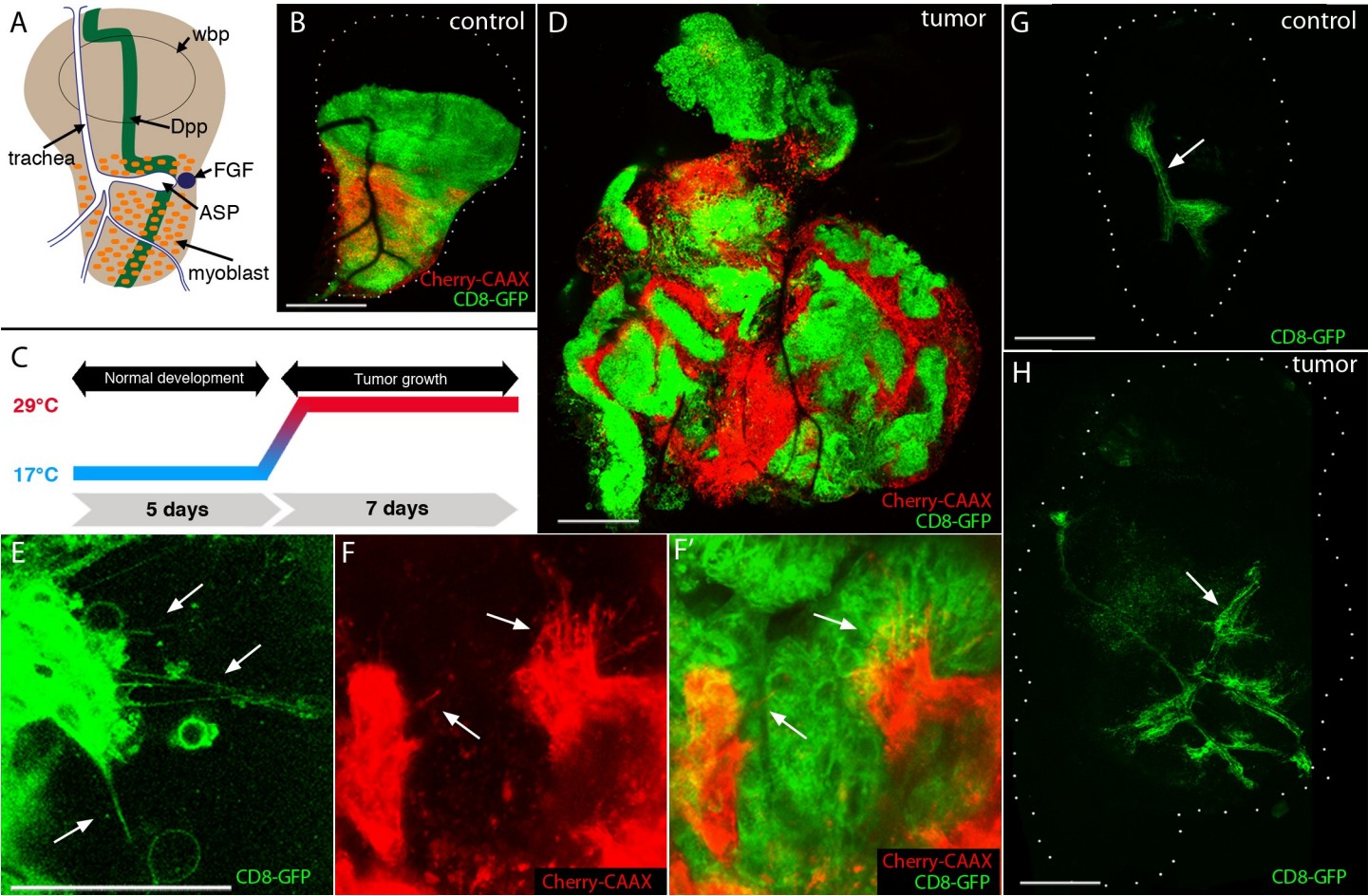

**Fig 1. Cytonemes in EGFR-Pcn$_i$ tumor and tumor-associated cells.** (A) Cartoon of a 3$^{rd}$ instar larval wing disc with wing blade primordia (wbp), disc-associated myoblasts (orange), trachea (white, outlined in blue) and air sac primordium (ASP), Dpp expressing cells (green stripe), Bnl expressing cells (blue circle). (B) Control wing disc expressing CD8:GFP in dorsal driven epithelial cells (*ap-Gal4*; green), and mCherry:CAAX in myoblasts ((*15B03-lexA*; red). (C) Schematic representation of tumor induction: animals developed for five days at 18˚C with Gal4 repressed by Gal80, were transferred to 29˚C to induce Gal4 and tumor growth for seven days (unless indicated otherwise). (D-F) Unfixed EGFR-Pcn tumor model wing discs. (D) Wing disc with tumor cells (CD8:GFP; green) and myoblasts (mCherry:CAAX; red). Genotype: *ap-Gal4,UAS-psq$_{RNAi}$/115B03-lexA,lexO-Cherry-CAAX;UAS-EGFR,tub-Gal80$^{ts}$/UAS-CD8:GFP*. Scale bars: 100µm. (E) Cytonemes in the tumor epithelial cells (green, arrows); genotype: *ap-Gal4,UAS-psq$_{RNAi}$/+;UAS-EGFR,tub-Gal80$^{ts}$/UAS-CD8:GFP*. (F-F') Cytonemes in myoblasts (red, arrows) extend towards epithelial cells (green); genotype: *ap-Gal4,UAS-psq$_{RNAi}$/115B03-lexA,lexO-Cherry-CAAX;UAS-EGFR,tub-Gal80$^{ts}$/UAS-CD8:GFP*; scale bars: 50µm. (G-H) Unfixed wing discs with marked tracheal cells (green, arrows); (G) control, genotype: *btl-LHG,lexO-CD2-GFP*; (H) EGFR-Pcn tumor, genotype: *ap-Gal4,UAS-psq$_{RNAi}$/btl-LHG,lexO-CD2-GFP; UAS-EGFR,tub-Gal80$^{ts}$/+*. Excessive tracheal growth and ectopic branches indicated by arrows; scale bars: 100µm.

with CD8:GFP. Dpp:mCherry fluorescence was present in the cytonemes of the epithelial tumor cells (Fig 2B, 2B' and 2B"), consistent with the possibility that Dpp signaling is mediated by cytonemes in the EGFR-Pcn tumor model.

## Genetic suppression of tumor phenotypes

To assess the role of cytonemes in tumorigenesis, we examined discs in which cytonemes are impaired. Downregulation of *Nrg, Caps, SCAR,* or *dia* decreases the number and length of cytonemes, and decreases signaling in tracheal cells, myoblasts and wing disc cells [7,9,18]. Nrg and Caps are cell adhesion proteins and SCAR and Dia are actin-binding proteins. Although severe loss-of-function conditions for *Nrg, caps, SCAR or dia* are lethal, the partial

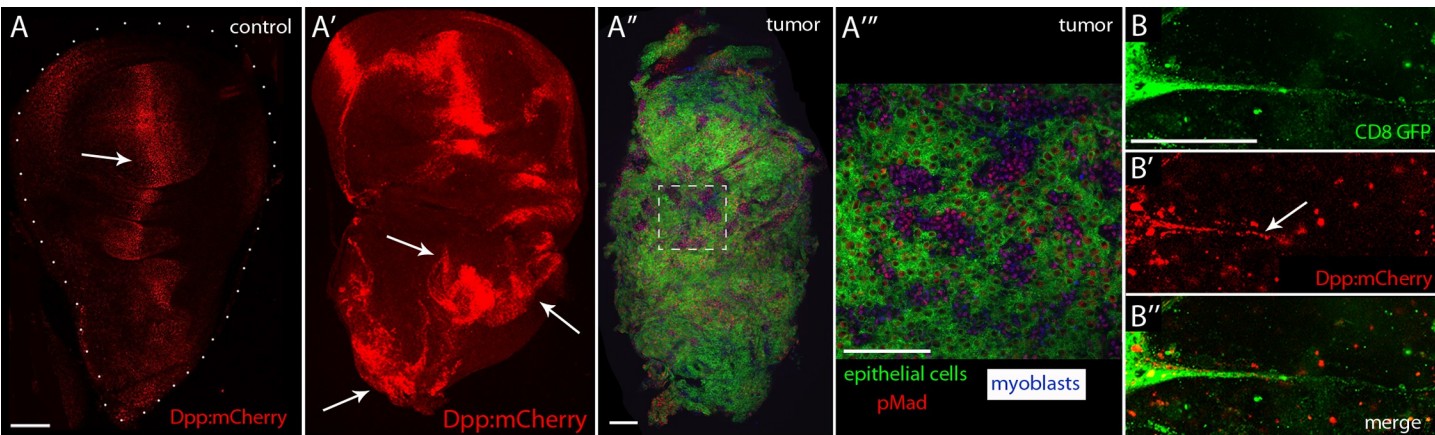

**Fig 2. Dpp signaling in the EGFR-Pcn tumor model.** (A,A',B) Unfixed wing discs showing the Dpp distribution. (A) Control disc with Dpp (red, arrow). Genotype: *Dpp: mCherry*. Scale bar: 100μm. (A') Disc with EGFR-Pcn tumor induced for 5 days expressing Dpp (red) in tumor cells. Genotype: *ap-Gal4,UAS-psq_{RNAi}/Dpp:mCherry; UAS-EGFR,tub-Gal80^{ts}/UAS-CD8:GFP*. Arrows indicate Dpp up-regulation. (A"-A"') EGFR-Pcn wing showing epithelial tumor cells (green) fixed and stained with α-phosphorylated MAD (pMad, red) antibody to monitor Dpp signaling and α-Cut (blue) to label myoblasts. Genotype: *ap-Gal4,UAS-psq_{RNAi}/+;UAS-EGFR,tub-Gal80^{ts}/ UAS-CD8:GFP*. Scale bar: 100μm. (A"') Higher magnification image of the box area in (A"), scale bar: 50μm. (B-B") Cytoneme (green) extending from epithelial tumor cell (green) with Dpp:mCherry (red); arrow indicates Dpp:mCherry in cytoneme. Genotype: *ap-Gal4,UAS-psq_{RNAi}/Dpp:mCherry;UAS-EGFR,tub-Gal80^{ts}/UAS-CD8:GFP*. Scale bar: 50μm.

loss-of-function conditions we used and previously characterized do not perturb cell polarity, cell viability, or cell cycle during normal development [7,9,19,20]. Previous studies of the wing disc and associated tracheal cells and myoblasts identified cytonemes that either "send" signaling proteins from producing cells or "receive" signaling proteins from target cells, and reported cytonemes that link disc cells to each other or to tracheal cells or myoblasts [9,18,20–22]. Available genetic tools can be used to impair cytoneme function but they do not distinguish among these types of cytonemes.

For the tumor discs with diminished Nrg, Caps, SCAR, or Dia, we compared disc morphology, Dpp signaling (monitored by anti-pMad antibody staining), and myoblast distribution (monitored by anti-Cut antibody staining, a marker of myoblasts [23]) in three types of wing discs: control non-tumor discs, EGFR-Pcn tumor discs and EGFR-Pcn tumor discs that also expressed Caps^{DN} or RNAi constructs targeting Nrg, Dia, or SCAR. These genotypes were generated from two crosses. In the first, EGFR-Pcn tumor discs were generated from a cross between *ap-Gal4,UAS-psq_{RNAi}/CyO;UAS-EGFR,tub-Gal80^{ts}* and *UAS-CD8:GFP* that produces equal numbers of animals with the tumor genotype and non-tumor controls that have the *CyO* balancer and lack *ap-Gal4,UAS-psq_{RNAi}*. The animals were incubated to the 2nd instar stage at low temperature (18˚C) to permit repression of the transgenes by Gal80^{ts} and were incubated at non-permissive temperature (29˚C) thereafter (Fig 1C). The *CyO* control animals develop to late 3rd instar within one day and eclose in approximately four days as curly wing adults. All remaining animals developed tumors and were developmentally-delayed, and were analyzed after seven days of culture at 29˚C. The second cross mated *ap-Gal4,UAS-psq_{RNAi}/CyO; UAS-EGFR,tub-Gal80^{ts}* to flies with the respective "tester chromosome" carrying *UAS-Caps^{DN}* or *UAS-RNAi*, and were incubated with the same regimen involving removal of *CyO* balancer adults. The remaining larvae had tumor phenotypes to varied degrees.

Tumor discs were misshapen and approximately 6.3 times larger than control discs, their number of Cut-expressing cells increased by four times, and their anti-pMad staining was not patterned normally (Fig 3A and 3B). In contrast, discs with tumor cells that expressed NrgRNAi in addition to EGFR and Pcn were morphologically less distorted, only 1.8 times

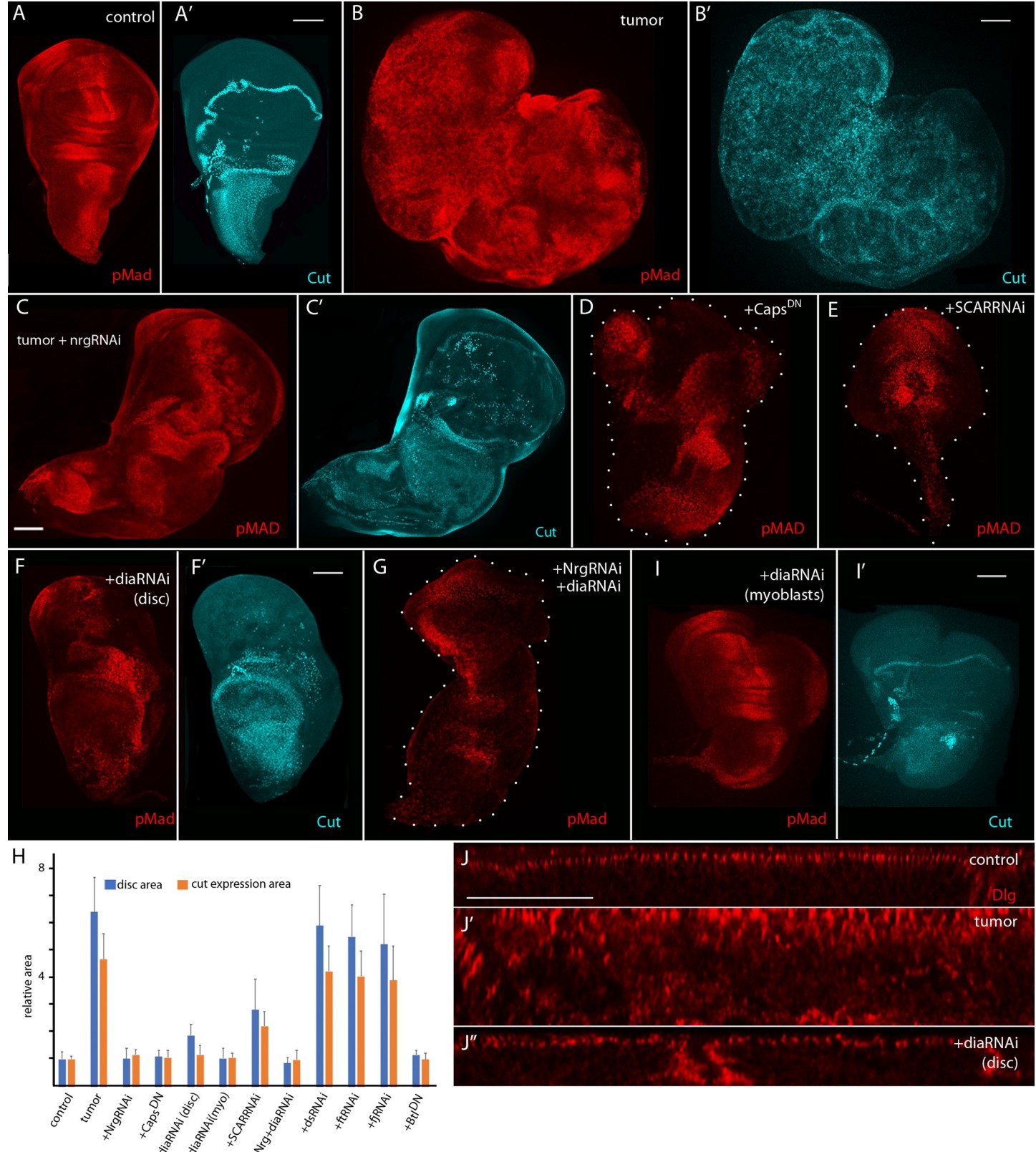

**Fig 3. Conditions that ablate cytonemes decrease signaling, reduce myoblast and tumor growth.** (A-I) Fixed wing discs stained with α-phosphorylated MAD (pMad, red) antibody to monitor Dpp signaling and α-Cut (cyan) to label myoblasts. Scale bars: 100μm. (A,A') Control. (B,B') EGFR-Pcn tumor, genotype: *ap-Gal4,UAS-psq_RNAi/ +;UAS-EGFR,tub-Gal80^ts/UAS-CD8:GFP*. (C,C') Tumor + NrgRNAi, genotype: *ap-Gal4,UAS-psq_RNAi/UAS-Nrg_RNAi;UAS-EGFR,tub-Gal80^ts/+*. (D) Tumor + Caps^DN, genotype:*ap-Gal4,UAS-psq_RNAi/UAS-CAPS^DN;UAS-EGFR,tub-Gal80^ts/+*. (E) Tumor + SCARRNAi, genotype:*ap-Gal4,UAS-psq_RNAi/+;UAS-EGFR,tub-Gal80^ts/ UAS-SCAR_RNAi*. (F,F') EGFR-Pcn + diaRNAi in epithelial cells. Genotype: *ap-Gal4,UAS-psq_RNAi/+;UAS-EGFR,tub-Gal80^ts/UAS-dia_RNAi*. (G) Tumor + NrgRNAi, diaRNAi, genotype: *ap-Gal4,UAS-psq_RNAi/UAS-Nrg_RNAi;UAS-EGFR,tub-Gal80^ts/UAS-dia_RNAi*. (I-I') EGFR-Pcn + diaRNAi expressed in the myoblasts. Genotype: *ap-Gal4,UAS-psq_RNAi/115B03-lexA;UAS-EGFR,tub-Gal80^ts/ lexO-dia_RNAi*. (H) Quantification of the total wing disc area (blue) and relative area of Cut-expressing cells (orange) of control, EGFR+Pcn tumor, tumor + diaRNAi expressed in epithelial cells, diaRNAi expressed in myoblasts, NrgRNAi, SCARRNAi, Caps^DN, NrgRNAi +diaRNAi, dsRNAi, ftRNAi, fjRNAi and Btl^DN larvae. Data was normalized to control. Student's *t* test *P* values (*P* >1.10^−9 for all except no significant difference for tumor + dsRNAi, ftRNAi and fjRNAi); *n* = 15–20 discs for each genotype. (J-J") Sagittal sections of fixed wing discs stained with α-Dlg antibody (red) to mark the cell's apical compartments, Scale bar: 50μm. (J) Control. (J') EGFR-Pcn tumor, genotype: *ap-Gal4,UAS-psq_RNAi/UAS-CD8:GFP;UAS-EGFR,tub-Gal80^ts/+)* (J") EGFR-Pcn + diaRNAi expressed in epithelial cells, genotype: *ap-Gal4,UAS-psq_RNAi/+;UAS-EGFR,tub-Gal80^ts/UAS-dia_RNAi*.

larger than controls, and the number and distribution of Cut-expressing cells was close to normal (Fig 3C and 3C'). In these animals, expression of EGFR, Pcn, and NrgRNAi is driven by *ap-Gal4* continuously after the second instar, but the noxious effects of NrgRNAi suppress the tumor phenotype induced by EGFR and Pcn overexpression and are tolerated by the disc cells in which NrgRNAi is expressed. The implication is that tumor cells are more sensitive to the consequences of Nrg downregulation than are normal cells. Hyper-sensitivity to sub-lethal levels of toxic conditions is a common hallmark of tumor cells.

Expression of CAPS^DN, SCARRNAi, or diaRNAi in the epithelial cells of the EGFR-Pcn model also reduced tumor growth, pMAD expression and number of Cut-expressing cells (Fig 3D–3F'). Expression of diaRNAi also suppressed excessive tracheation in the tumor discs (Fig 3S). To test whether the suppressive, ameliorative effects might be additive, we expressed NrgRNAi and diaRNAi simultaneously in EGFR-Pcn tumor cells. We did not observe that the degree of tumor suppression changed relative to expression of either NrgRNAi or diaRNAi alone (Fig 3G). We also tested the roles of three genes that are essential for planar cell polarity [24]: the *dachsous* (*ds*) and *fat (ft)* genes that encode cadherin family proteins, and *four-jointed* (*fj*) that encodes a transmembrane kinase. Expression of dsRNAi, ftRNAi or fjRNAi does not perturb cytoneme-mediated signaling between wing disc and tracheal cells [19], and expression of these RNAi lines in the tumor cells had no apparent effect on tumorigenesis (S2 Fig). Fig 3H summarizes the growth suppression we observed in the genetic conditions we tested.

The presence of cytonemes in both the tumor columnar epithelial and mesenchymal myoblast cells, and the essential role of the myoblasts for tumor progression raises the possibility that myoblast cytonemes might also play an essential role in tumorigenesis. To investigate the role of myoblast cytonemes, we expressed diaRNAi (with *1151-lexA lexO-diaRNAi*) in the myoblasts of discs that overexpress EGFR and Pcn in the columnar epithelial cells. The morphology, Dpp signaling pattern and myoblast growth characteristic of the EGFR-Pcn tumors were suppressed (Fig 3I and 3I'). This result is consistent with the idea that the myoblasts signal to the epithelial tumor cells [12], and that this signaling is mediated by cytonemes.

We also analyzed the apical-basal organization of the disc cells by monitoring the distribution of Discs large (Dlg), which associates with the septate junction and localizes to the apical compartment of the columnar epithelial cells. Sagittal optical sections of discs stained with anti-Dlg antibody revealed that the specific apical distribution of Dlg characteristic of wild type cells is disorganized in EGFR-Pcn tumor discs (Fig 3J and 3J'). Expression of diaRNAi in the tumor cells restored the Dlg distribution to normal (Fig 3J"). This demonstrates that expression of diaRNAi suppresses a critical feature of tumor cells and that downregulation of Dia is compatible with normal cellular morphology and behavior.

Although EGFR and Pcn expression in the EGFR-Pcn model (driven by *ap*-Gal4) is restricted to the dorsal compartment of the wing disc, the tumors grow extensively and metastasize (Fig 4A). The tumorous condition is 100% lethal; animals with these tumors do not

mature beyond the larval stage [12]. However, the conditions of Nrg, Caps, SCAR, or dia downregulation that suppress tumor growth also suppressed lethality: the number of EGFR-Pcn tumor-bearing larvae that pupated and that reached the pharate adult stage increased, and for the animals that expressed diaRNAi, approximately 60% survived to adult stage (Fig 4B). These surviving adults were fertile, and wing blade morphological defects were the only visible phenotype (Fig 4C and 4D). Given that cytoneme-mediated signaling is reduced by downregulation of Nrg, Caps, SCAR, or Dia, these results are consistent with the possibility that cytoneme-mediated signaling is necessary for tumor growth and that interfering with signaling either between tumor cells or between tumor and stromal cells suppresses many if not all aspects of tumorigenesis.

## Bnl signaling in the EGFR-Pcn model

The disc-associated ASP branch of the tracheal system is dependent on and sensitive to signals produced by the disc [25,26], and Bnl signaling from the disc to the ASP is cytoneme-mediated and cytoneme-dependent [7,21]. Because tracheal branches grow excessively in the EGFR-Pcn tumor model (Fig 1H), we investigated if Bnl signaling is upregulated in tumor discs. Bnl is normally produced by a small, discrete group of disc cells (Fig 1A). Disc cells do not express Btl, but tracheal cells express Btl and not Bnl [26]. To monitor Bnl signaling in the EGFR-Pcn

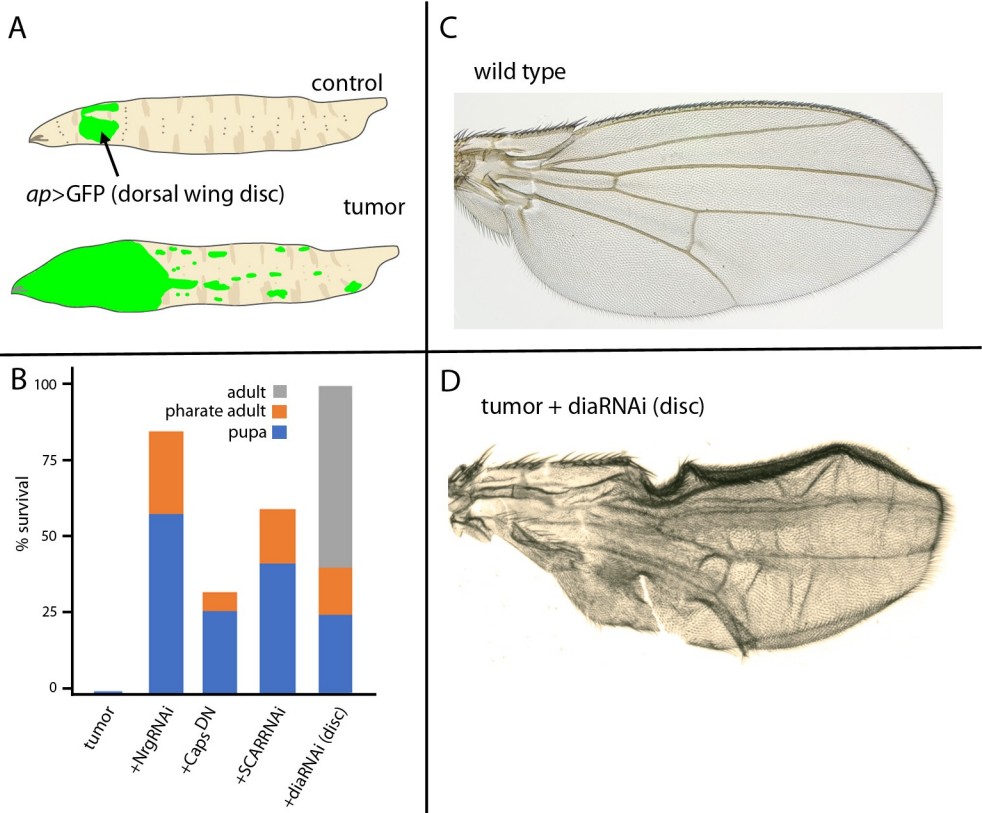

**Fig 4. Conditions that ablate cytonemes promote survival.** (A) Cartoon of a wild type larva depicting cells expressing GFP in the imaginal disc dorsal compartments (*ap*-Gal4; green, arrow) and of EGFR-Pcn tumor larva with overgrowth and metastasis throughout the larva (green). (B) Survival of EGFR-Pcn tumor, tumor + diaRNAi, NrgRNAi, SCARRNAi and Caps$^{DN}$ larvae to pupal (blue), pharate adult (orange) and adult stage (gray). Student's *t* test *P* values: (between $P<0.05$ and $P>1.10^{-8}$) with $n$ = 15–30 larvae for each genotype. (C) Control adult wing. (D) EGFR-Pcn + diaRNAi wing, genotype: *ap-Gal4,UAS-psq$_{RNAi}$/UAS-CD8:GFP;UAS-EGFR,tub-Gal80$^{ts}$/UAS-dia$_{RNAi}$*.

tumor model, we examined a Bnl reporter that expresses mCherry:CAAX in Bnl-expressing cells [27]. The number and location of Bnl-expressing cells increased in tumor discs (Fig 5A and 5B). We also examined fluorescence of Btl:mCherry (with a CRISPR-generated knock-in [21]). Whereas Btl:mCherry fluorescence was not detected in the epithelial cells of normal wing discs (Fig 5C and 5C'), Btl:mCherry fluorescence was present in many epithelial cells of the tumor (Fig 5D and 5D'). These results are consistent with the possibility that the tumor induces ectopic expression of Btl and that ectopic activation of the Bnl signaling pathway might correlate with excessive growth of the tracheal branches in this tumor.

To investigate the role of Bnl signaling in EGFR-Pcn tumorigenesis, we overexpressed a dominant negative FGFR mutant in the tumor cells to block Bnl signaling ($UAS$-$Btl^{DN}$). We monitored wing discs for morphology, Cut expression, and pMAD in EGFR-Pcn tumor discs, and EGFR-Pcn tumor discs that also express Btl$^{DN}$. Experimental crosses were carried out with the regimen described previously and produced either tumor larvae or suppressed tumor larvae; both crosses generate control ($CyO$ balancer) and tumor-containing animals in a Mendelian ratio of 1:1. In the cross with $UAS$-$Btl^{DN}$, 50.6% (76/151) of the larvae pupated and eclosed within 4 days as curly wing adults. The presence of the balancer chromosome indicates that the genotype of these flies lacked $ap$-$Gal4$, as expected of control, non-tumor animals. The remaining larvae do not develop beyond the pupal stage, consistent with their having the tumor genotype. Larvae analyzed after 7 days of Gal4 expression at the non-permissive were compared to tumor discs of the same age (Fig 1C). In the EGFR-Pcn tumor discs that also express Btl$^{DN}$, characteristics of tumor morphology, size, pattern of Dpp signaling, and distribution of myoblasts were suppressed (Fig 5E and 5F).

To confirm the identity genotype of the suppressed tumor discs, RNA isolated from wild type, EGFR-Pcn tumor, and Btl$^{DN}$-expressing tumor discs was quantified by QPCR. This analysis confirmed the overexpression of EGFR in both tumor and suppressed tumor discs (Fig 5G). We also examined EGF and Bnl signal transduction in tumor and suppressed tumor discs by staining with anti-dpERK antibody. The presence of dpERK was observed in control, tumor, and Btl$^{DN}$-over-expressing control and tumor discs, and whereas the pattern of dpERK in the tumor discs was expanded and disordered in the tumor discs, the patterns and levels in the suppressed discs was close to normal (Fig 5H–5K). These findings are consistent with the idea that tracheogenesis is necessary for tumor growth and with a previous report that describes comparable findings in studies of a *lethal giant larvae* Drosophila tumor model [28]. In this tumor, ectopic tracheal sprouting is associated with hypoxic responses and tracheal differential of wing disc tumor cells, a process that may be analogous to "sprouting angiogenesis" and vascular co-option in mammalian tumors [29].

## Cytonemes in a RET-MEN2 tumor model

We investigated the role of cytonemes in the Drosophila RET-MEN2 tumor model developed by the Cagan lab [14]. This model mimics the mis-regulation of signaling pathways that have been implicated in MEN2-related tumors. Overexpression of RET$^{MEN2}$ in a discrete set of wing disc epithelial cells (with *ptc-Gal4*) resulted in a >4X increase in the number of *ptc*-expressing cells and a 7X increase in the portion of the disc that consists of *ptc*-expressing cells (Fig 6A and 6B) [14]. Approximately one-half of the animals survive to the pupal stage, but none survive to adult. We tested whether expression of Irk2$^{DN}$ (an inwardly-rectifying potassium channel required for cyteneme-mediated signaling [5]), diaRNAi, or SCARRNAi in the RET-mutant cells affects tumor growth and survival. We observed that excessive growth of the *ptc*-expressing cells was suppressed by more than 2X in all three genotypes (Fig 6C–6F). Approximately two-thirds of the animals developed to the pupal stage, and survival to adult

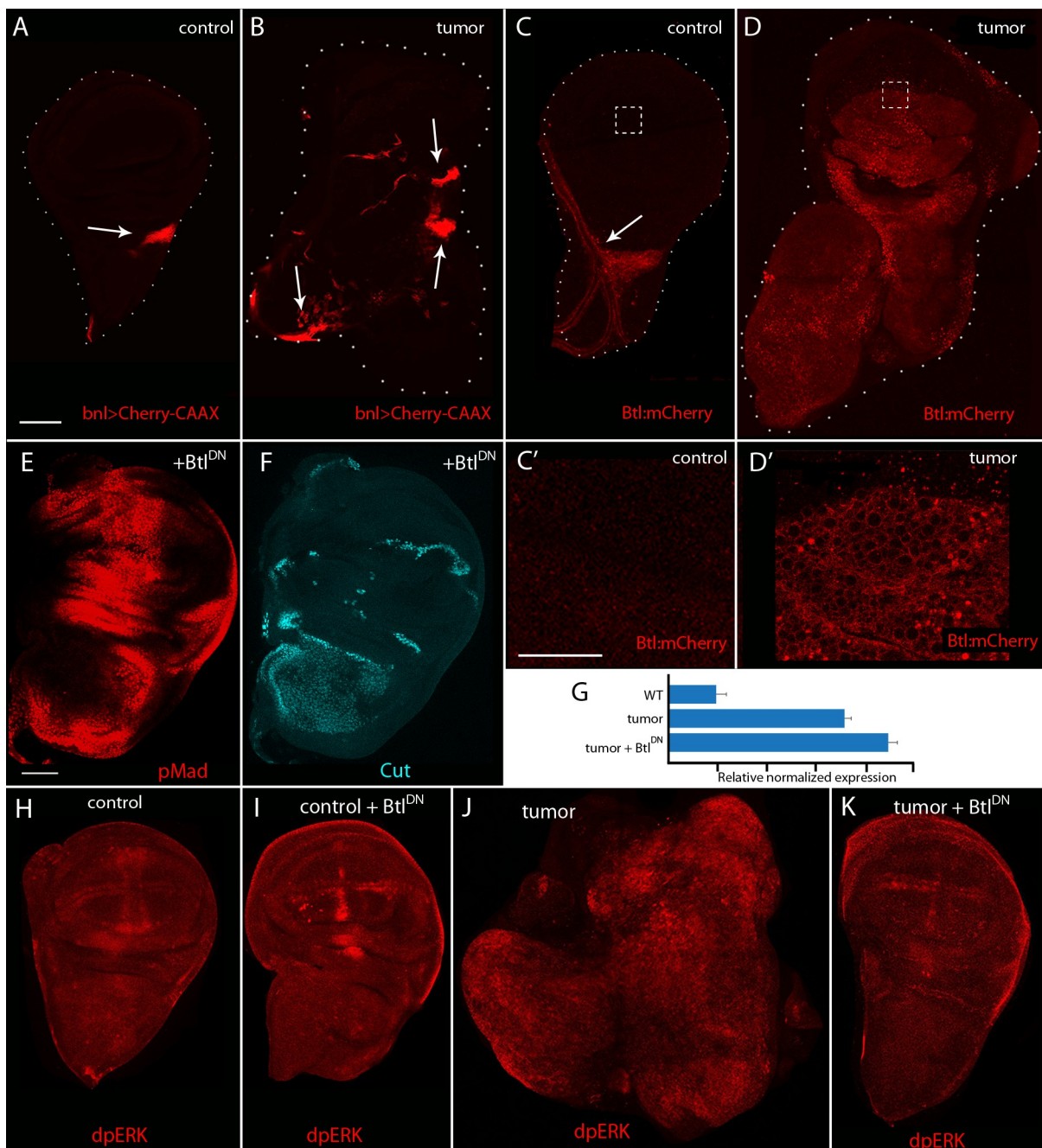

**Fig 5. FGF signaling in the EGFR-Pcn tumor model.** (A-B) Unfixed wing discs with Bnl-expressing cells marked with Cherry-CAAX (red, arrows). (A) Control, genotype: *bnl-lexA,lexO-mCherry:CAAX/+*. Scale bar: 100μm. (B) EGFR-Pcn tumor, genotype: *ap-Gal4,UAS-psq_{RNAi}/+; UAS-EGFR,tub-Gal80^{ts}/bnl-lexA,lexO-mCherry:CAAX*. Arrows indicate FGF up-regulation (red). (C-D') Unfixed wing discs with Btl distribution marked by Btl:mCherry (red, arrows). (C) Control, genotype: *Btl:mCherry/+*. Btl is only expressed in the tracheal cells. Scale bar: 100μm (C') Higher magnification image of the boxed area in (C). (D) 5 day EGFR-Pcn tumor disc, genotype: *ap-Gal4,UAS-psq_{RNAi}/ UAS-CD4-mIFP;UAS-EGFR,tub-Gal80^{ts}/Btl:mCherry*. Btl expression is upregulated in the tumor cells. (D') Higher magnification image of the box area in (D). Scale bars: 50μm. (E-F) EGFR-Pcn tumor +Btl^{DN} fixed wing disc stained with α-phosphorylated MAD (pMad, red) antibody to monitor Dpp signaling (E) and α-Cut (cyan) to label myoblasts (F). Scale bar: 100μm. Genotype: *ap-Gal4,UAS-psq_{RNAi}/+;UAS-EGFR,tub-Gal80^{ts}/UAS-Btl^{DN}*. (G) EGFR mRNA measured by quantitative real-time PCR. RNA was extracted from 5 discs of the indicated genotypes. Data were normalized to rp49. The data show mean ± SD from three technical replicates of a representative experiment. Significance was analyzed using Student's t-test (p<0.001). Comparable results were obtained in 3 independent biological replicates. Genotypes are *ap-Gal4, UAS-CD8:GFP* (WT), *ap-Gal4,UAS-psq_{RNAi}/+;UAS-EGFR,tub-Gal80^{ts}/ UAS-CD8:GFP* (tumor) and *ap-Gal4,UAS-psq_{RNAi}/+;UAS-EGFR,tub-Gal80^{ts}/UAS-Btl^{DN}*. (H-K) Fixed wing disc stained with α-phosphorylated ERK to monitor EGF signaling (dpERK, red). (H) Control,

genotype: *ap-Gal4,UAS-CD8:GFP*. (I) Control+Btl$^{DN}$, genotype: *ap-Gal4/+;UAS-Btl$^{DN}$/+*. (I) EGFR-Pcn tumor disc, genotype: *ap-Gal4,UAS-psq$_{RNAi}$/+;UAS-EGFR,tub-Gal80$^{ts}$/ UAS-CD8:GFP*. EGFR-Pcn tumor + Btl$^{DN}$, genotype: *ap-Gal4,UAS-psq$_{RNAi}$/+;UAS-EGFR,tub-Gal80$^{ts}$/ UAS-Btl$^{DN}$*.

also increased (Fig 6G). These flies have normal morphology, and with the exception of small wing vein abnormalities, the wings are indistinguishable from wild type (Fig 6H). These results are consistent with a general role for cytonemes in tumorigenesis and tumor progression.

## Discussion

The tumor microenvironment is a niche that responds to signaling proteins produced by tumor cells and supplies growth factors that support tumor growth and metastasis [30]. Much ongoing work seeks inhibitors of tumorigenesis that target the signaling molecules and growth factors, their signal transduction pathways, and the stromal cells of the microenvironment [31,32]. Two previous studies reported cellular extensions of human tumor cells in ex vivo co-cultures with non-metastatic cells and in vivo, and have been implicated these structures in material transfer between tumor and non-tumor cells [33,34]. In this work, we also investigated the mechanism that transfers signaling molecules and growth factors between tumor cells and stromal cells in vivo, and report the first evidence for their essential role in tumorigenesis.

Previous work established that during Drosophila development, paracrine signaling by the signaling proteins/growth factors Dpp, Bnl, Wg, Notch and Hedgehog, is mediated by cytonemes [35–37]. Cytonemes are specialized filopodia that extend between signal producing and signal receiving cells, making synaptic contacts where the signaling proteins transfer from producing to receiving cells. To extend this work to tumorigenesis, we applied the strategies and tools we developed for previous studies to ask if cytonemes are present in the tumor microenvironment, and if genetic conditions that inhibit cytoneme function and cytoneme-dependent signaling in normal development also inhibit tumorigenesis. In a EFGR-Pcn tumor model, we found that cytonemes extend from both Drosophila tumor and stromal cells (Fig 1). This is consistent with previous studies that reported increased signaling between tumor and stromal cells in this model [12], and with the presence of cytonemes in many other contexts of paracrine signaling [7,18,20,38–41]. We confirmed that Dpp is expressed by the tumor cells (Fig 2; [12]), and found that ectopic Bnl signaling also has an essential role in this tumor (Fig 5). These results imply functional connections between the EGF, Dpp, and Bnl signaling pathways in this tumor, and although we did not identify regulatory interactions between the pathways, our results show that ectopic activation of the Bnl pathway is essential to tumorigenesis.

We also found conditions that impair cytonemes and rescue flies of lethal tumors in both EGFR-Pcn and RET models. We selected five genes from among the more than twenty that are known to be essential for cytoneme-mediated signaling [5,7,18,19]. *nrg*, *caps*, *Irk2*, *SCAR*, and *dia* are recessive lethal genes whose functions can be partially reduced in genetic mosaics without affecting viability, cell shape, or the cell cycle, but are necessary for cytoneme function. Downregulating any one of these genes improved viability in the tumor models. *dia* downregulation is the most effective inhibitor of cytoneme-mediated signaling in other contexts [7,19,20], and it is the most effective in both tumor models. The cures that downregulation effected suggest that cytoneme-mediated signaling, which might be a general mechanism for tumorigenesis in a variety of cancers, might also be a potential target for therapy.

The high degree of evolutionary conservation of Drosophila and human proteins makes Drosophila a clinically relevant platform for understanding mechanisms human disease, and Drosophila tumor models have successfully identified new therapeutic candidates for

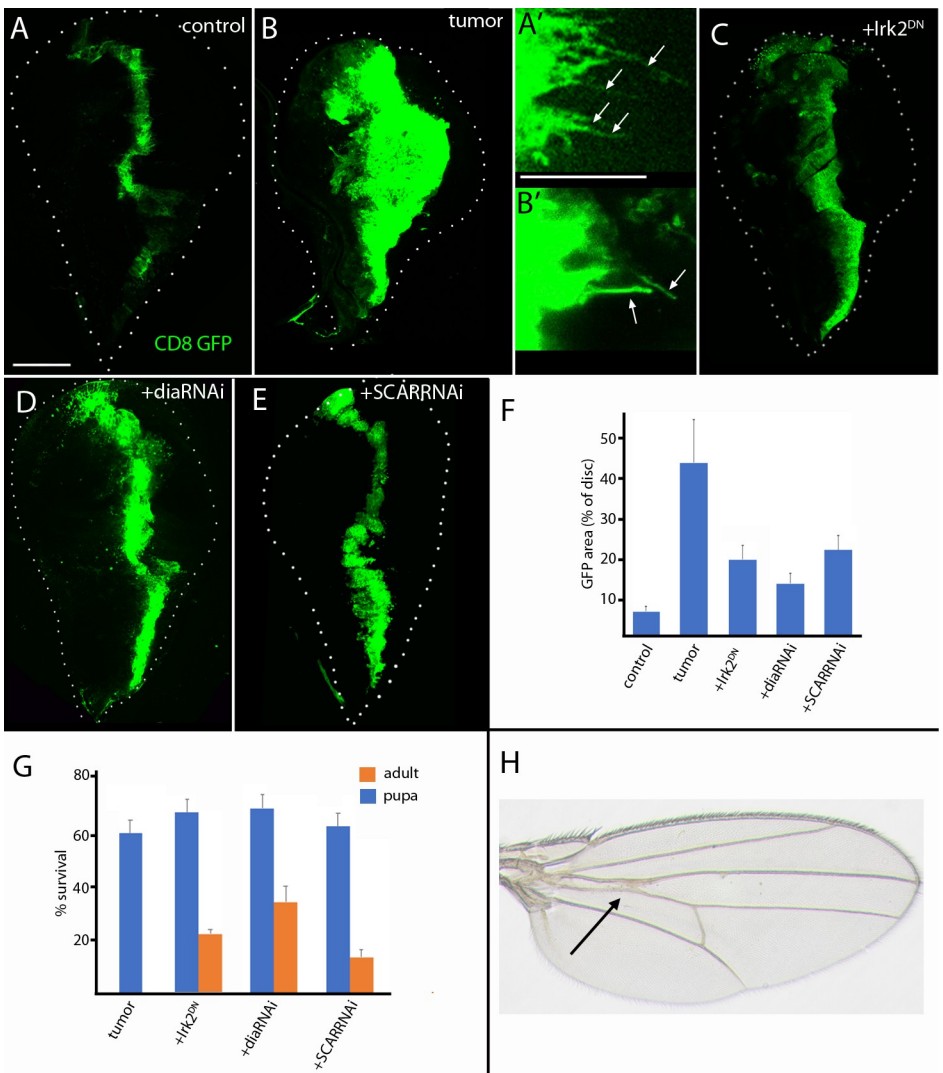

**Fig 6. RET-MEN2 tumor growth and survival depends on cytonemes.** (A-B') Unfixed wing discs expressing CD8:
GFP (green) driven by *ptc-Gal4*. Scale bar: 100μm. (A) Control. (A') Cytonemes in wild type cells (green, arrows). Scale
bar: 50μm. (B) RET tumor, genotype: *ptc-Gal4,CD8:GFP;UAS-RET$^{MEN2}$/UAS-CD8:GFP*. (B') Cytonemes in RET
tumor cells (green, arrows). (C-E) Unfixed wing discs expressing CD8:GFP (green) and either RET + diaRNAi (C);
RET + SCARRNAi (D); and RET + Irk2$^{DN}$ (E). (C) Genotype: *ptc-Gal4,CD8:GFP;UAS-RET$^{MEN2}$/UAS-dia$_{RNAi}$*. (D)
Genotype: *ptc-Gal4,CD8:GFP;UAS-RET$^{MEN2}$/UAS-SCAR$_{RNAi}$*. (E) Genotype: *ptc-Gal4,CD8:GFP/Irk2$^{DN}$;
UAS-RET$^{MEN2}$*. (F) Quantification of the area of the disc expressing GFP (% of disc) in control, RET tumor, or RET
and either diaRNAi, SCARRNAi or Irk2$^{DN}$. Significance was analyzed using Student's *t*-test ($P<1.10^{-5}$) with 15–18
discs. (G) Survival of RET-tumor and RET and either diaRNAi, SCARRNAi or Irk2$^{DN}$ to pupa (blue) and adult
(orange). Significance using Student's *t*-test for adults is ($P<0.001$), $n = 30$–40 for each genotype. (H) RET tumor,
diaRNAi wing blade; genotype: *ptc-Gal4,CD8:GFP;UAS-RET$^{MEN2}$/UAS-dia$_{RNAi}$*. Arrow points to abnormal crossvein.

colorectal, lung and thyroid and stem-cells derived cancers [42–44]. Our work provides proof
principle for tumor suppression by interfering with cytoneme-mediated signaling.

## Materials and methods

### Drosophila stocks and husbandry

Flies were reared on standard cornmeal and agar medium at 29˚C, unless otherwise stated. *ap*-
Gal4 UAS-psq$_{RNAi}$/CyO; UAS-EGFR *tub*-Gal80$^{ts}$ from S. Cohen [12], UAS-RET$^{MEN2}$ from R.

Cagan [14], Btl:mCherry and Bnl-lexA, from S. Roy [21,27], lexO-dia$_{RNAi}$ from H. Huang, UAS-Caps$^{DN}$ [45] (deletion mutant lacking the intracellular domain), UAS-Btl$^{DN}$ from B. Shilo [46] (dominant negative construct lacking a functional cytoplasmic tyrosine-kinase domain), Irk2$^{DN}$ from E. Bates [47] (a subunit predicted to block the channel). *btl-LHG,lex-O-CD2:GFP*, a tracheal-specific driver [16]; *ptc*-Gal4 enhancer is an enhancer trap line that mimics *ptc* expression [48], lexO-mCherry:CAAX from K. Basler; lines from Bloomington Stock Center: *15B03*-lexA (#52486), UAS-CD8:GFP (#5137), UAS-dia$_{RNAi}$ (#28541 and #35479), UAS-NrgRNAi (#37496), UAS-ds$_{RNAi}$ (#32964), UAS-ft$_{RNAi}$ (#34970), UAS-fj$_{RNAi}$ (#34323); and UAS-SCAR$_{RNAi}$ (#21908) from Vienna Drosophila Research Center Stock Center.

## Dpp:mCherry

The Dpp:mCherry transgene has mCherry inserted C-terminal to Dpp amino acid 465 [17], with Leu-Val linkers inserted before and after a mCherry coding sequence deleted of its stop codon. The transgene was generated by CRISPR mutagenesis as follows:

## Dpp:mCherry donor vector

Left homology arm fragment contains overlapping sequence with PBS-SK vector and mCherry. The mCherry fragment contains overlapping sequence with the left homology arm and right homology arm. The right homology arm fragment contains overlapping sequence with mCherry and PBS-SK vector. The three fragments were stitched together and cloned into PBS-SK vector using Gibson Assembly (NEB). The resulting vector is designated as Dpp: Cherry donor vector.

Left arm homology sequence was amplified from wild-type genomic DNA using:

L-arm-fwd: cggtatcgataagcttgatcaccttgccgcacaaatacatatac

L-arm-rev: CCTCGCCCTTGCTCACCATCTCCAGGCCACCGCCCTCTCCGGCAGCACGTCCCGA

The mCherry tag was amplified using:

mCherry-fwd:TGTCTGCCGGAGAGGGCGGTGGCCTGGAGATGGTGAGCAAGGGCGAGGAGGATAAC

Cherry-rev:CGCTTGTTCCGGCCGCCCTTCTCTAACTTGTACAGCTCGTCCATGCCGC

The right arm homology sequence was amplified from wild-type genomic DNA using:

R-arm-fwd:GGACGAGCTGTACAAGTTAGAGAAGGGCGGCCGGAACAAGCGGCAGCCGA

R-arm-rev:ccgggctgcaggaattcgatGTCATTATTCGGTTATGCTCTCGCTAG

pCFD-3 gRNA vector

gRNA sequence: CGCTCCATTCGGGACGTGTCTGG

The gRNA sequence without the PAM was cloned into pCFD-3 vector obtained from Addgene.

## Dpp:Cherry CISPR lines

pCFD-3 gRNA vector and Dpp:mCherry donor vector were co-injected into Cas9 expressing flies (nanos-Cas9) by Rainbow Transgenics. The resulting CRISPR-generated flies were screened and verified by sequencing. The Dpp:mCherry homozygous fly is viable and has normal morphology. The distribution of Cherry fluorescence in the wing disc is consistent with images in [17,49], and the gradient of Cherry fluorescence in the columnar epithelial cells of the disc is intracellular (S1 Fig).

## EGFR-Pcn tumor

EGFR-Pcn tumors were induced as described by Herranz et al, [12] by overexpression of EGFR and down-regulation of *pipsqueak* (*psq*), which leads to increased levels of Pcn. Female flies from the stock *ap-Gal4,UAS-psq$_{RNAi}$/CyO;UAS-EGFR,tub-Gal80$^{ts}$* were crossed to males of the corresponding genotypes at 18°C, and were cultured at 18°C to maintain Gal80 repression of Gal4 and allow normal development. After 5 days larvae were transferred to 29°C to induce Gal4 expression and tumor growth. 4 days after the temperature shift CyO/+ flies eclosed and were removed from the vial. Tumor growth was induced for 7 days, unless otherwise indicated, whereupon larvae were dissected for live imaging or immunostaining, or were maintained at 29°C for survival studies. To control for possible effects on Gal4 expression, all tested genotypes had three UAS transgenes–either UAS-EGFR, UAS-psqRNAi and UAS-CD8: GFP for tumor flies, or UAS-EGFR, UAS-psqRNAi and additional RNAi for comparisons. Experimental and control crosses were carried out in parallel.

## RET tumor model

Female flies from the RET$^{MEN2}$ stock [14] were crossed at room temperature to either *ptc-Gal4*, 2xUAS-CD8:GFP males or either with UAS-diaRNAi, or SCARRNAi or Irk2$^{DN}$ males. For analysis of discs, embryos from one day collections were transferred to 29°C and cultured to third instar stage. For survival comparisons, animals were cultured at 25°C.

## Live imaging of wing imaginal discs

Wing discs with trachea attached were dissected in cold phosphate-buffered saline (PBS), placed on a coverslip and mounted upside-down on a coverslip on a depression slide as described [9]. Samples were imaged with a Leica TCS SPE confocal or an Olympus FV3000 inverted confocal laser scanning microscope.

## Immunohistochemistry

Wing discs were dissected in cold PBS and fixed in 4% formaldehyde for 20 minutes. After extensive washing, the samples were permeablized with PBST (PBS + 0.3% TritonX-100), blocked for 1h with PBST+3%BSA blocking buffer, and incubated with primary antibodies previously diluted in blocking buffer overnight at 4°C. The following primary antibodies were used: α-pMad (Abcam), α-Discs large (Dlg), α-Cut and α-β-galactosidase (Developmental Studies Hybridoma Bank). Secondary antibodies were conjugated to Alexa Fluor 405, 488, 555, or 647. Samples were mounted in Vectashield and imaged with a Leica TCS SPE confocal or an Olympus FV3000 inverted confocal laser scanning microscope.

## Image analysis and quantification

All measurements and quantifications of wing discs were done in z-section stacks of confocal images using Fiji software from 15–20 discs for each genotype. Total wing disc area and Cut-expressing cells in the EGFR-Pcn tumor or GFP-expressing cells in the RET tumor, were quantified by measuring the mean intensity of fluorescence relative to the total area of the wing disc. Data was normalized to control. Statistical significance values were calculated with Student's *t* test.

## Quantitative real-time PCR

Total RNA was extracted from 5 wing discs of either wild type, EGFR-Pcn tumor or EGFR-Pcn tumor + BtlDN larvae using the RNeasy Micro Kit (Quiagen). Larvae corresponding from 3

genotypes were under the same temperature conditions (5 days of tumor induction at 29˚C). Reverse transcription was carried out using the Applied Biosystem High Capacity RNA-to-cDNA. qPCR reactions were performed with a BioRad C1000 Touch Thermal Cycler and SYBR Green (Bioline). qPCR results were analyzed according to the comparative threshold cycle (Ct) method, where the amount of target, normalized to an endogenous actin reference and relative to an experimental control, is given by 2−ΔΔCt. Ct represents the PCR cycle number at which the amount of target reaches a fixed threshold. The ΔCt value is determined by subtracting the reference Ct value (rp49) from the target Ct value. ΔCt was calculated by subtracting the ΔCt experimental control value.

## Supporting information

**S1 Fig. Dpp:mCherry distribution in the wing disc.** Wing discs from a L3 larva with the CRISPR-generated Dpp:mCherry allele stained with phalloidin (green) to mark the cells. Frontal section shows Cherry fluorescence extending anteriorly and posteriorly from the band of Dpp expression. Orthogonal section shows that the Cherry fluorescence in the cells outside the band of Dpp expression is intracellular.
(TIF)

**S2 Fig. Conditions that don't ablate cytonemes do not reduce tumor growth.** (A) Unfixed wing disc with marked tracheal cells (green). Genotype: *ap-Gal4,UAS-psq$_{RNAi}$/btl-LHG,lexO-CD2-GFP;UAS-EGFR,tub-Gal80$^{ts}$/UAS-diaRNAi*. (B-D) Fixed wing discs stained with α-pMad (red) antibody to monitor Dpp signaling and α-Cut (cyan) to label myoblasts. Scale bar: 100μm. (B-B') Tumor + fjRNAi, genotype: *ap-Gal4,UAS-psq$^{RNAi}$/+;UAS-EGFR,tub-Gal80$^{ts}$/UAS-fj$_{RNAi}$.* (C-C') Tumor + dsRNAi, genotype: *ap-Gal4,UAS-psq$^{RNAi}$/+;UAS-EGFR,tub-Gal80$^{ts}$/UAS-ds$_{RNAi}$.* (D-D') Tumor + ftRNAi, genotype: *ap-Gal4,UAS-psq$^{RNAi}$/+;UAS-EGFR, tub-Gal80$^{ts}$/UAS-ft$_{RNAi}$.*
(TIF)

## Acknowledgments

We thank: Dr. Manolo Calleja for initiating this study, Drs. S. Cohen, H. Herranz, S. Roy, and R. Cagan and Bloomington Stock Center for fly stocks and R. Cagan for his help and advice; all members of Kornberg lab for discussion and constructive suggestions.

## Author Contributions

**Conceptualization:** Sol Fereres, Ryo Hatori, Thomas B. Kornberg.

**Data curation:** Sol Fereres.

**Funding acquisition:** Thomas B. Kornberg.

**Investigation:** Sol Fereres, Ryo Hatori.

**Methodology:** Sol Fereres, Ryo Hatori.

**Project administration:** Thomas B. Kornberg.

**Resources:** Makiko Hatori.

**Supervision:** Thomas B. Kornberg.

**Validation:** Sol Fereres.

**Visualization:** Sol Fereres, Ryo Hatori.

Writing – original draft: Sol Fereres, Thomas B. Kornberg.

Writing – review & editing: Sol Fereres, Thomas B. Kornberg.

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
