## [Decision Letter · Decision Letter 0]

2 Sep 2019

[EXSCINDED]

Dear Tom:

We finally got the reviews for your Research Article entitled 'Cytoneme-mediated signaling essential for tumorigenesis' to PLOS Genetics. As we only had very partial reviews from PLOS Biology, we had to send the manuscript back for review and we managed to obtain three in depth reviews.

As you will see, all three reviewers feel that the work is important and should be published. However, two of them ask for clarifications for several points and I hope that you will be able to answer these points without taking too much time. One of the reviewers asks for further experiments, but hopefully, you should be able to argue the points and explain in a rebuttal how you would like to address this.

I hope that you will be able to get back to us very shortly with a revised document.

We therefore ask you to modify the manuscript according to the review recommendations before we can consider your manuscript for acceptance. Your revisions should address the specific points made by each reviewer.

[LINK]

Cordially,

Claude

Claude Desplan

Associate Editor

PLOS Genetics

Gregory Barsh

Editor-in-Chief

PLOS Genetics

Reviewer's Responses to Questions

**Comments to the Authors:**

**Reviewer #1:** 

In this revised manuscript, the authors examine the role of cytonemes in two cancer models, EGFR/Perlecan and Ret. This work builds on previous work from the Kornberg laboratory demonstrating the central role of cytonemes in a broad array of cell-cell signaling paradigms. The data is clean and clearly presented, and the authors have addressed the previous reviewers' issues. The impact of demonstrating that cytonemes signal in cancer is potentially large, as the field is seeing an increasing appreciation of the importance of local epithelial biology on tumor progression and therapeutic response.

One challenge of this work is cleanly removing cytonemes without affecting other aspect of the cells' biology. The authors argue that showing requirement for a diverse set of six genes linked to cytonemes is the same as demonstrating a requirement for cytonemes in transformation. The problem with this argument is that each of the six loci is important for a central aspect of cell biology, and their requirement for transformation is not surprising. However, the Kornberg laboratory has a long track record of demonstrating a role for cytonemes. Evidence for cytoneme involvement in transformation of these fly cancer models includes:

• the authors show extensive cytoneme networks associated with the tumors

• the presence of cytonemes is closely correlated with activity of EGFR etc. based on previous work

• genes that disrupt cytonemes consistently reduce aspects of transformation; the Kornberg laboratory has previously extensively characterized their partial loss-of-function phenotypes, which is different from aspects of cell transformation

• the work demonstrating a role for Bnl provides strong evidence for cytoneme involvement, since the requirement for cytoneme-based signaling by Bnl is well established.

The authors have strong evidence for the role of cytonemes in transformation. However, they should note in the manuscript that their data is "consistent with a role for cytonemes in tumor progression" but that they cannot rule out that these six loci also alter tumor progression through their other known cell biology roles. A more conservative description better captures the data.

Small points:

• The authors use the word “cure” at multiple points throughout the manuscript. A subset of animals are rescued to adulthood by preventing transformation through genetic or drug manipulation prior to transformation. This is not a cure, and is not the same as reversing transformation with a drug, a requirement in clinical studies to claim a cure.

• The authors state that emergent Ret adults have no phenotype when the activity of cytoneme regulators is reduced. Given the use of the patched driver, they should show in the wing or elsewhere that, for example, the wing patterning is normal. This would be a rigorous test for the requirement of the different loci or drugs.

• Figure 5S should be moved to Supplemental data.

**Reviewer #2: **

The manuscript by Fereres et al. examine the role of cytonemes in two Drosophila tumor models. The main conclusion is that cytonemes are necessary for tumor growth as they promote signaling between epithelia cells and mesenchymal cells/myoblasts. The study is of interest to PLoS Genetics readers, however, the model they present is suggestive and not conclusive. More experiments are needed to make the argument more compelling.

A key observation is that in the tumor model (overexpression of wild type EGFR and perlecan), Bnl/FGF and its receptor Btl are ectopically expressed, and that blocking Btl RTK signaling in epithelial cells overexpressing EGFR + perlecan suppresses proliferation. The authors conclude, as Bnl/Btl signaling has previously been implicated in cytoneme formation, "that cytonemes traffic the signaling proteins that move between tumor and stromal cells.” An alternative model is that Btl signaling in epithelial cells upregulates ERK signaling (in synergy with overexpressed EGFR) independently of cytonemes. The authors should activate MAPK signaling, which can be done in a number of ways, in the absence of Btl in epithelial cells and examine whether epithelial cells can proliferate and whether they have or do not have cytonemes. If they have cytonemes then they should remove them using DiaRNAi (for example) and test whether cytonemes are required for activated MAPK proliferation. If cytonemes are not required for proliferation then one would conclude that ERK activation is all the epithelial cells need for proliferation which would put in question the conclusion that Btl role in epithelial cells is to build cytonemes that in turn are required to signal to myoblasts to receive in turn growth factors.

The authors propose that signaling from myoblasts through myoblast cytonemes is important for proliferation of epithelial tumors. I presume that it is Wnt signaling from the myoblasts that is important. It would be of interest to express Wntless RNAi (with 1151-lex, lexO-Wintless RNAi) in the myoblasts of discs that overexpress EGFR + Pcn in the columnar epithelial cells, to clarify this issue.

The authors propose that "These results suggest that tumor growth might be dependent on ectopic tracheation”. There is no data in the manuscript to test the role of trachea on epithelial cells proliferation The only data is the observation that there are more trachea in discs overexpressing wild type EGFR and perlecan, which is not surprising as they produce more Bnl.

Data on the Ret tumor model are very preliminary and do not add much to the study.

**Reviewer #3:**

In "Cytoneme-mediated signaling essential for tumorigenesis", Fereres et al present the first evidence of the role of signaling specialized cell protrusions (cytonemes) during tumorigenesis. This manuscript approaches the mechanisms for cell-to-cell communication in tumor formation; presenting striking data for the inhibition of tumor growth by the abrogation of cytoneme formation in Drosophila experimental models. To do this, the authors use regulated partial inhibition of proteins required for the formation of cytonemes linked to Dpp and FGF pathways. Cytoneme mediated transport of these signals between wing disc and stroma cells (myoblasts and ASP) seems to be important for tumor development.

As a whole I consider that the manuscript presents important results supporting the crucial role of cytoneme-mediated signaling in the intercellular communication during tumor development. It embodies the advances in the application of ground-breaking research regarding cell communication within a disease scenario. Therefore, I recommend its publication, however, there are some important issues that should be addressed first, mainly regarding clarity in the model proposed as well as in the actual experimental evidence for cytoneme-mediated signaling on tumor growth.

Major concerns:

1. In general, the manuscript lacks clarity regarding the attempted parallelisms between the Drosophila experimental models used and a classic tumor system. Through the manuscript it would be useful if authors kept defining the signaling of tumor versus stroma tissues. In the current text version is not always clear whether interpretations are refereeing to cytoneme-mediated signaling between wing disc and myoblast (or ASP) cells or among wing disc cells and actual tumor tissue overgrowth.

2. The experiment showing requirement of cytoneme-mediated signaling from myoblasts for wing disc tumor formation is very striking and would imply an absolute requirement of stromal cell-signaling for tumor growth. However, further experimental evidence is necessary regarding the signaling implicated, as rescue of the tumor growth when cytonemes are abrogated in the wing disc epithelium is also presented. In general characterization of cytonemes in tumor, no-tumor and rescued conditions are lacking, and quantifications and/or directionality analysis for the different situations would further support the cytonemes crucial role.

3. There are no experimental evidences showing the possible effect in cell death after the expression of the cytoneme abrogating RNAis. The tumor rescue could be due to tumor cell death induced by the expression of those tools.

4. Although the authors present strong evidence for increased FGF signaling during tumorigenesis, as well as impressive rescue evidence upon its inhibition, no data is presented towards the effects of induced ectopic FGF signaling in the wing disc. Would over-expression of Bnl or Btl in the wing disc be sufficient to induce tumor growth? Would this ectopic expression increase cytoneme occurrence? if there is already published evidence regarding these aspects, authors should at least refer it. In the same line, experimental evidence for FGF signaling effects upon cytoneme abrogation (as presented for Dpp signaling) would further clarify cytoneme function over tumor growth processes.

5. A final schematic model figure could be very useful and further explain data interpretation about stroma and tumor interaction; this figure should also include signal dependence between both tissues for tumor growth.

Minor concerns:

1. Methods for imaging analysis and quantifications should be included.

2. Regarding the use of the Dpp-mCherry insertion and as there has been some controversial publications (Entchev et al., 2000) arguing the potential loss of tags upon ligand processing (Harmansa et al., 2015); it would be desirable to include tests proving the tagging of the expressed protein in these flies (for example through an anti-Cherry WB).

3. The presentation of images showing Dpp-Cherry on cytonemes from tumor discs could benefit from the inclusion of information such as moving direction.

4. Figure 5G further labeling would help interpretation. As for the figure presented in between Figure 5 and 6 (Figure 5S) there is no labeling at all at present, and it should at least show the names of markers used.

5. Images presenting Ptc expressing cells in Fig. 6 should be labeled regarding the fluorescent reporter used (CD8GFP?)

6. Quantification of tumor disc area upon conditions that do not ablate cytonemes (Supplementary Figure 3) should be presented.

7. I might have missed something but after careful manuscript reading I found that the only genetic inhibition that rescues to adulthood is that of Dia and not the other cytoneme induced factors. Thus, a brief discussion of author’s interpretation towards this difference should be included.

**Have all data underlying the figures and results presented in the manuscript been provided?**

Reviewer #1: Yes

Reviewer #2: None

Reviewer #3: Yes

PLOS authors have the option to publish the peer review history of their article (what does this mean?). If published, this will include your full peer review and any attached files.

Reviewer #1: Yes: Ross Cagan

Reviewer #2: No

Reviewer #3: No

---

## [Editor Report · Decision Letter 1]

11 Sep 2019

Dear Tom,

We are pleased to inform you that your manuscript entitled "Cytoneme-mediated signaling essential for tumorigenesis" has been editorially accepted for publication in PLOS Genetics. Congratulations! and I hope that the long process was not too painful!

Cordially,

Claude,

Claude Desplan

Associate Editor

PLOS Genetics

Gregory Barsh

Editor-in-Chief

PLOS Genetics

Comments from the reviewers (if applicable):

Data Deposition

http://datadryad.org/submit?journalID=pgenetics&manu=PGENETICS-D-19-01310R1

Press Queries

---

## [Editor Report · Acceptance letter]

23 Sep 2019

PGENETICS-D-19-01310R1 

Cytoneme-mediated signaling essential for tumorigenesis 

Dear Dr Kornberg, 

We are pleased to inform you that your manuscript entitled "Cytoneme-mediated signaling essential for tumorigenesis" has been formally accepted for publication in PLOS Genetics! Your manuscript is now with our production department and you will be notified of the publication date in due course.

With kind regards,

Kaitlin Butler

PLOS Genetics

On behalf of:
